# Texture and Flexural Fatigue Resistance Governed by Surface-Dependent Deformation and Recrystallization in the Copper Foils

**DOI:** 10.3390/nano16010011

**Published:** 2025-12-20

**Authors:** Tong Wu, Guohao Liu, Di Liu, Bingxing Wang, Bin Wang, Yong Tian

**Affiliations:** State Key Laboratory of Digital Steel, Northeastern University, Shenyang 110819, China; wut6@mails.neu.edu.cn (T.W.); liugh7@mails.neu.edu.cn (G.L.); liud7@mails.neu.edu.cn (D.L.); wangbin@ral.neu.edu.cn (B.W.); tianyong@ral.neu.edu.cn (Y.T.)

**Keywords:** flexural fatigue resistance, grain-size control, recrystallization behavior, cube texture, stacked rolling

## Abstract

High-flexibility copper foils are critical for reliable flexible interconnects and displays. In this work, commercial-purity copper belts were processed by triple-layer stacked cold rolling to ultrathin foils, producing distinct surface- and layer-dependent deformation structures in the bright, matte, and central-interface layers; subsequent annealing at 600 °C then promoted orientation-selective recrystallization. Under the present conditions, the center-interface layer of the triple-rolled foil achieved the highest flexural-fatigue life (≈8.0 × 10^4^ cycles) within a window of cube ≈ 30–45% and grain size ≈ 40–60 μm. In this regime, grain-size control stabilizes intergranular slip compatibility, reduces elastic–plastic mismatch, and mitigates strain localization during cyclic bending. Even without aggressive cube enrichment, high flexural fatigue resistance can likewise be achieved through deliberate control of grain size. These findings establish a clear processing–microstructure–property linkage and indicate that layer-dependent control of texture and grain size can enhance flexural-fatigue performance in triple-layer stacked-rolled copper foils for flexible electronics.

## 1. Introduction

Copper foils are extensively employed as conductive layers in printed circuit boards (PCBs) and flexible printed circuits (FPCs), playing a critical role in communication, packaging, and emerging flexible electronic devices. With the increasing demand for miniaturized and durable components, improving the bending fatigue resistance of rolled copper foils has become a key requirement for ensuring long-term reliability [1,2]. Understanding how the microstructure and grain size evolve during rolling and subsequent annealing is therefore essential for optimizing processing parameters and enhancing flexural fatigue resistance through precise structural control [3,4,5].

Rolling deformation is the principal fabrication process for ultra-thin copper foils, as it provides high efficiency and strong control over both microstructure and texture. Severe plastic deformation during rolling introduces dense dislocation networks and cell structures, supplying the stored energy required for subsequent recrystallization. The through-thickness distribution of stored energy is closely linked to the development of deformation texture components (s, brass, copper) and in turn influences grain-growth behavior during annealing [6,7]. Liu et al. demonstrated that double rolling can generate a gradient microstructure along the thickness direction, characterized by spatial variations in grain size and dislocation density that significantly improve fatigue resistance by delaying crack initiation and propagation [8]. Similarly, Li et al. reported that a double-cross rolling strategy produces a comparable thickness-dependent gradient structure in copper foils, where enhanced dislocation storage and strain partitioning promote more uniform recrystallization and improved mechanical uniformity [9]. These findings highlight the importance of grain size heterogeneity and microstructural gradients in improving the fatigue performance of copper foils. However, most prior studies have focused on single-layer or double-layer routes; the capacity of triple-layer stacked rolling (TR), combined with calibrated annealing, to simultaneously stabilize the cube fraction and establish an operable grain-size window for enhanced flexural-fatigue life (N_f)_ has not been systematically quantified.

Annealing plays a critical role in tailoring the final properties of rolled copper foils. During recovery and recrystallization, stored energy is progressively released, enabling the nucleation and growth of strain-free grains. Cantergiani et al. showed that texture evolution during annealing is strongly conditioned by the prior processing path, which sets the stored-energy distribution and the propensity for orientation-selective recrystallization (cube development) [10]. Dong et al. found that appropriately tuned annealing yields uniform recrystallization and stable grain size, whereas miscontrolled schedules induce over-softening and non-uniform coarsening that degrade mechanical performance [11]. Accordingly, optimizing the annealing schedule to obtain a uniform recrystallized microstructure with controlled grain size and stable texture is important for improving flexural fatigue resistance [12].

In this study, copper foils produced by triple-layer stacked rolling were annealed at 600 °C for different durations. We measured and analyzed changes in grain size, recrystallization indicators, and texture and examined how grain-size changes together with cube evolution relate to flexural fatigue life, with the aim of providing a practical route to optimize the flexural fatigue resistance of rolled copper foils for flexible electronics.

## 2. Materials and Methods

### 2.1. Material

Commercial-purity copper (99.99 wt%) belts with dimensions of 200 mm × 50 mm × 0.15 mm were used as the starting material. The foils were fabricated using a straight-pull four-high reversible cold-rolling mill operating at a speed of 0.03 m s^−1^ under dry conditions, as illustrated in Figure 1.

Three copper foils were stacked together and rolled simultaneously to a final single-foil thickness of 0.024 mm, corresponding to a thickness reduction of approximately 84.5% and ensuring identical total reduction and processing conditions for each layer.

After rolling, the triple-rolled foils were annealed in a tubular furnace under a flowing argon atmosphere at 600 °C for 7.5, 15, 30, 60, and 120 min, followed by immediate water quenching to preserve the annealed microstructure.

For clarity, the sample notation used throughout this study follows the format “TR–Y–Z”, where

Y = layer position (C: center layer, O: outer layer);

Z = surface type (B: bright, M: matte).

Accordingly, the analyzed specimens include TR-C-M (center interface surface), TR-O-B (outer bright surface), and TR-O-M (outer matte surface).

### 2.2. Cyclic Bending Tests

The flexural fatigue life (N_f_) of the foils was evaluated using a cyclic bending test, representing high-cycle bending fatigue. Rectangular specimens with dimensions of 12.7 mm × 100 mm were prepared and mounted on an FPC fatigue testing machine, as schematically illustrated in Figure 2. During testing, the foil was bent over a fixed radius of 2.5 mm with a stroke length of 25 mm at a frequency of 175 cycles min^−1^. Both ends of the copper foil were electrically connected, and the test was continued until an interruption of the current (open-circuit event) was detected and a visually observable fracture of the foil was confirmed; this combined event was taken as the failure criterion. For each condition, five specimens were tested. The flexural fatigue life was defined as the arithmetic mean of the numbers of cycles to failure for all five specimens in each condition. The raw fatigue-life data are provided in Appendix A.

Cyclic bending tests were performed on three types of foils, denoted TR-O-B, TR-O-M, and TR-C-M. For TR-O-B and TR-O-M specimens, the bright surface (TR-O-B) or matte surface (TR-O-M) was placed on the inner side of the bending radius, and the opposite side corresponded to the interface surface. For TR-C-M specimens, both surfaces exhibit similar interface-type microstructures and experience nearly symmetric stress states during bending; therefore, no distinction is made between the two surfaces of TR-C-M in the present study.

### 2.3. Tensile Tests

Tensile tests were carried out at room temperature, using a universal testing machine (CMT5105, Shenzhen SANS Testing Machine Co., Ltd., Shenzhen, China) with a speed of 0.1 mm/min. For uniaxial tensile tests, specimens were grouped as TR-C (center foil) and TR-O (outer foils). The gauge length and width of tensile samples were 12 mm and 4 mm. For data accuracy, the samples were tested in triplicate.

### 2.4. Microstructural Characterization

Microstructural characterization was performed using a field-emission scanning electron microscope (GeminiSEM 560, Carl Zeiss Microscopy GmbH, Oberkochen, Germany) equipped with EBSD. EBSD measurements were performed on both cold-rolled and annealed specimens using different magnifications and step sizes optimized for grain-size and texture statistics. EBSD measurements were performed on three types of foils obtained from the triple-rolled laminate, denoted TR-O-B, TR-O-M, and TR-C-M. For TR-O-B, mappings were acquired on the outer bright surface; for TR-O-M, on the outer matte surface; and for TR-C-M, on the surface of the central foil that had been located at the internal interface in the stacked configuration. All EBSD mappings were carried out on foil surfaces (plan-view). For the cold-rolled condition, high-magnification maps (≈3000×, step size 0.18 μm) were acquired for grain-size analysis, whereas lower-magnification maps (≈500×, step size 0.4 μm) were used to obtain representative orientation distribution functions and texture components. For the annealed foils, large-area maps were recorded at a magnification of ≈100× with a step size of 2 μm in order to include a large number of grains (typically on the order of several thousand for each condition) and ensure that the measured texture and grain-boundary statistics are statistically representative. EBSD data were acquired and analyzed using AZtecCrystal software (v2.1). A critical misorientation angle of 15° was applied to identify grain boundaries. Low-angle grain boundaries (LAGBs) were defined within the range of 2–15°, and high-angle grain boundaries (HAGBs) as those above 15°. The analysis focused on grain size, grain boundary character distribution, kernel average misorientation (KAM), and geometrically necessary dislocation (GND) density. Texture characterization was carried out using inverse pole figure (IPF) maps, texture component maps, and orientation distribution function (ODF) sections, with a maximum deviation angle of 15° applied to calculate the volume fractions of cube, copper, s, goss, and brass components.

TEM samples were prepared from the central interface layer (TR-C-M) in both the cold-rolled and annealed conditions. TEM samples were mechanically pre-polished and then punched into 3 mm diameter disks. The disks were subsequently thinned using an argon ion milling system until perforation occurred. The ion milling was conducted at an incident angle of 7° and an accelerating voltage of 4 kV. The prepared specimens were examined using a field-emission transmission electron microscope (JEM-F200, JEOL Ltd., Akishima, Tokyo, Japan) operated at 200 kV.

### 2.5. Evaluation of Orientation-Averaged Young’s Modulus from ODFs

Orientation-averaged Young’s modulus along the rolling/bending direction (E_RD_) was evaluated from the measured ODFs. For each orientation g specified by Bunge Euler angles, the corresponding orientation matrix was used to obtain the direction cosines (l_1_, l_2_, l_3_) of the rolling direction in the crystal frame. The single-crystal modulus E_k_ along this direction was calculated from the cubic compliance constants of copper, S_11_, S_12_, S_44_, using:
(1)1Ek=S11−2Q(l12l22+l22l32+l32l12),Q=S11−S12−S442

For each ODF grid point, the measured orientation density f_k_ and the metric factor sinΦ_k_ were combined into a normalized weight ω_k_:
(2)ωk=fksinΦk∑jfjsinΦj

The Voigt and Reuss averages of the directional modulus were then computed as
(3)EV=∑kωkEk,1ER=∑kωkEk

The effective Young’s modulus along the rolling direction was taken as the Hill mean:
(4)ERD=(EV+ER)2

## 3. Results

### 3.1. Microstructural and Texture Evolution During Rolling and Annealing

Figure 3 presents the EBSD inverse pole figure (IPF) maps, grain-size distributions, and kernel average misorientation (KAM) maps of cold-rolled triple-rolled (TR) copper foils at different layer positions. As shown in Figure 3a–c, all layers exhibit grains markedly elongated along the rolling direction. The TR-C-M layer shows more slender and directionally aligned grains, whereas TR-O-B is relatively coarser with weaker elongation [13].

The grain-size distribution is positively skewed, dominated by sub-micrometer grains (<1 µm) with a modest right tail toward larger sizes. TR-C-M shows the highest fraction of <1 µm grains and the narrowest tail. In TR-O-B and TR-O-M, the <1 µm fraction is slightly reduced and the 1–3 µm tail modestly enhanced, yielding averages of ~1.63 µm (TR-O-B) and ~1.62 µm (TR-O-M).

The KAM maps (Figure 3g–i) reveal layer-dependent differences in local misorientation. TR-C-M shows a larger share of higher-KAM regions (see legend), indicating stronger local lattice curvature; TR-O-B exhibits overall lower KAM levels; TR-O-M is intermediate between the two. The corresponding GND densities are on the order of 10.05 × 10^14^ m^−2^ for TR-O-B, 11.22 × 10^14^ m^−2^ for TR-O-M, and 12.74 × 10^14^ m^−2^ for TR-C-M. Surface-dependent differences in GND density reflect through-thickness plastic strain gradients, which are associated with the elastic energy stored in the dislocation stress field; therefore, the systematic differences in KAM and GND density between TR-O-B and TR-C-M indicate a clear through-thickness gradient in deformation heterogeneity across the three layers.

Figure 4 shows the evolution of (a) high-angle grain boundary (HAGB) fraction and (b) average grain size of triple-rolled Cu foils with annealing time.

As shown in Figure 4a, all layers exhibit a rapid increase in HAGB fraction within the first 7.5 min. With continued annealing, the HAGB fraction then decreases in all layers: the outer layers (TR-O-B and TR-O-M) decline slowly and remain above 90%, whereas TR-C-M drops most rapidly, falling below 30% at 120 min, because the post-annealing grain orientations become more strongly clustered, and the orientation contrast between neighboring grains is reduced. As a result, the intergranular misorientation angles shift toward lower values, leading to a decreased fraction of HAGBs [14].

As shown in Figure 4b, the average grain size of all layers increases steadily during annealing. The outer layers (TR-O-B and TR-O-M) maintain relatively fine and stable grains (<100 µm), whereas the center layer (TR-C-M) undergoes pronounced coarsening, reaching ~520 µm at 120 min. This accelerated coarsening in TR-C-M reflects that, in the cold-rolled state, this layer combines the highest stored strain energy with a deformation texture richer in S- and copper-type components, which provides both a stronger driving force and more favorable interfaces for cube nucleation and growth. As a consequence, TR-C-M develops the highest cube volume fraction after recrystallization, and cube grains can grow preferentially at the expense of the surrounding deformation-related orientations, while grain growth in the outer layers remains much less pronounced [10].

Figure 5 presents the orientation distribution functions (ODFs) at φ_2_ = 45° and 65° for triple-rolled Cu foils in the rolled and annealed states.

In the rolled condition (Figure 5a,b), all layers exhibit typical deformation textures dominated by the s, copper, and brass components [15], with relative orientation densities varying across layers. The TR-C-M layer shows the most pronounced copper component (MAX = 25.1 mrd) together with a strong s component (17.6 mrd), whereas the outer layers (TR-O-B and TR-O-M) display slightly lower deformation components, reflecting through-thickness strain heterogeneity introduced by triple stacking.

After short annealing at 600 °C for 7.5 min (Figure 5c,d), the deformation components weaken markedly and cube-oriented grains begin to appear, indicating the onset of recrystallization. A distinct cube peak (MAX ≈ 34.6 mrd) develops in TR-C-M.

Following annealing for 120 min (Figure 5e,f), the deformation components are largely replaced by a cube-dominated texture. The maximum cube orientation densities reach 39.6 mrd for TR-O-M and 51.4 mrd for TR-C-M, indicating substantial cube strengthening with prolonged annealing.

Figure 6 presents the distribution maps of the major texture components—cube, copper, s, goss, and brass—for triple-rolled Cu foils in the rolled and annealed states.

In the rolled condition (Figure 6a–c), all layers exhibit pronounced deformation textures dominated by the s and copper components, while brass and goss are sparse and discontinuous along elongated grain bands. Texture component fractions vary across the thickness: the center layer (TR-C-M) shows the highest fractions of s and copper; the outer bright surface (TR-O-B) displays comparatively lower fractions; and the matte surface (TR-O-M) is intermediate. These differences indicate a distinct through-thickness gradient introduced by triple stacking [16].

After annealing at 600 °C × 7.5 min (Figure 6d–f), all layers show partially recrystallized structures in which cube-oriented grains emerge within the deformation matrix. TR-C-M exhibits a markedly higher cube fraction (~38%), whereas the outer layers (TR-O-B and TR-O-M) retain larger fractions of deformation components, suggesting slower recrystallization progress.

With prolonged annealing for 120 min (Figure 6g–i), the microstructure evolves toward a cube-dominated configuration. TR-C-M attains a high cube fraction, while the outer layers preserve small fractions of copper and s.

Figure 7 displays the area fractions of the texture components (cube, copper, s, goss, and brass) on the three surfaces of the triple-rolled Cu foil during annealing. The center interface (TR-C-M) shows the fastest shift from deformation-related components to a cube-dominated state: the cube fraction rises steeply—approaching ~95% by 60 min—and then stabilizes, while the fractions of the s and copper textures drop to low levels. In contrast, both outer surfaces (TR-O-B and TR-O-M) evolve more gradually and non-monotonically: the cube fraction increases rapidly within the first 15 min, undergoes a temporary reduction between 15 and 60 min—attributable to orientation redistribution and competitive growth with deformation-related components—and rises again with prolonged annealing [17].

Figure 8 shows the evolution of the average grain size for different texture components and the corresponding orientation maps of the triple-rolled Cu foil after annealing at 600 °C for 15–60 min.

The center interface surface (TR-C-M) exhibits a distinctly accelerated coarsening behavior, manifested by the rapid expansion of cube-oriented grains and the concurrent disappearance of deformation-related orientations. In contrast, the outer bright and matte surfaces (TR-O-B and TR-O-M) display a pronounced orientation-dependent evolution. During annealing, the cube-oriented grains show a progressive decrease in mean size; from 30 to 60 min, the mean continues to decline, likely because numerous newly formed small cube grains emerge and depress the average. By comparison, deformation-related orientations such as s and brass exhibit transient coarsening between 15 and 30 min, followed by gradual shrinkage at longer durations, suggesting a temporary competitive growth advantage of the deformation textures before cube-dominated growth resumes. Overall, this non-monotonic trend reflects competition in grain-growth kinetics among different texture components [18,19].

### 3.2. Correlation Between Microstructure and Flexural Fatigue Resistance

Figure 9 illustrates the evolution of the flexural fatigue life (N_f_) of triple-rolled Cu foils as a function of annealing time and its correlation with the cube fraction and grain size.

The central interface surface (TR-C-M) shows a rapid increase in N_f_, reaching a maximum of about 8.0 × 10^4^ cycles at 7.5 min, followed by a sharp decline with further annealing. This non-monotonic behavior arises from the transition from moderate recrystallization to abnormal cube-dominated grain growth, where excessive coarsening reduces the intergranular constraint [20].

In contrast, the outer bright and matte surfaces (TR-O-B and TR-O-M) exhibit a gradual increase and subsequent stabilization of N_f_ with annealing. The improvement in flexural fatigue resistance corresponds to the progressive formation of cube texture and moderate grain coarsening, which together promote stable slip compatibility during cyclic bending [21].

As shown in Figure 9b, N_f_ rises with increasing cube fraction alongside grain size and is highest, in this study, for cube ≈ 30–45% combined with grain size ≈ 40–60 µm. Beyond this range, abnormal grain growth or insufficient cube formation leads to a marked decrease in fatigue life, suggesting that flexural fatigue resistance is jointly governed by cube texture development and controlled grain coarsening [22,23].

Figure 10 shows bright-field (BF) TEM images of TR-C-M specimens before and after cyclic bending. In the cold-rolled condition, the microstructure displays closely spaced dislocation cells and subgrains, with dense dislocation tangling in many regions. After annealing, recrystallized grains are larger and the images show fewer dense dislocation tangles in the pre-bending state. After cyclic bending, fresh slip traces and intragranular dislocation lines appear within individual grains [24].

Figure 11 shows the tensile stress–strain curves of the triple-rolled Cu foils annealed at 600 °C for different durations.

As shown in Figure 11a, the TR-C foil exhibits the highest strength in the rolled and short-annealed states, together with relatively limited elongation. This behavior is consistent with its smaller average grain size and high dislocation density at this stage, where the increased grain-boundary area and stored defects provide pronounced Hall–Petch-type strengthening. With further annealing, recovery and recrystallization reduce both grain-boundary and dislocation densities, leading to a marked decrease in strength and a concurrent increase in elongation [25]. The TR-O foil in Figure 11b shows a similar evolution, and its mechanical response becomes nearly stable after 30 min of annealing. Overall, annealing rapidly softens the foils while enhancing ductility, with TR-C maintaining a higher strength level at a given annealing time, and TR-O exhibiting slightly better elongation after full recrystallization.

## 4. Discussion

### 4.1. Layer-Dependent Deformation and Stored-Energy Partitioning

During cold rolling, layer-dependent grain refinement is governed by the stress–strain fields established at the roll–foil and foil–foil interfaces. The central interface surface (TR-C-M) experiences bidirectional constraint, which drives intense through-thickness strain transfer and compatibility stresses. These conditions promote dense dislocation tangling and pronounced lattice curvature, producing finer (~1.25 µm) and more elongated grains that are consistent with a highly strained deformation structure [26]. On the outer bright surface (TR-O-B), deformation approaches a plane-strain compression state because the stacked neighboring layer constrains lateral strain and lowers interfacial shear. This condition is expected to reduce the propensity for shear-related slip activation, leading to more uniform deformation and a moderate grain-size refinement (~1.62 µm) with limited grain elongation [27,28]. The outer matte surface (TR-O-M) experiences stronger interfacial shear and friction-induced strain localization. The imposed strain is accommodated by multidirectional dislocation rearrangement and lattice distortion, yielding nearly equiaxed grains of ~1.6 µm. This morphology is characteristic of shear-affected deformation modes in multilayer laminates [29].

These observations indicate that the triple-layer configuration introduces distinct deformation modes across its surfaces—predominantly plane-strain compression on the bright surface, enhanced interfacial shear on the matte surface, and constrained through-thickness strain at the center—forming a characteristic strain gradient.

### 4.2. Mechanisms of Cube Texture Evolution During Annealing

During annealing, the evolution of cube texture is governed by the interplay between stored-energy-driven nucleation and orientation-selective grain growth, both of which are conditioned by the strain heterogeneity introduced during rolling. Cube nuclei are often reported to form within high-dislocation-density transition bands adjacent to S- and copper-oriented regions, where local gradients in stored energy are sufficient to stabilize new orientations. In these locations, cube–matrix pairs frequently exhibit a misorientation of about 40° around <111>. Under the available driving pressure, such <111> high-angle boundaries possess relatively high mobility, promoting subgrain coalescence and boundary migration compared with other misorientation pairings. As a result, cube grains experience orientation-selective growth, and the overall texture gradually evolves toward cube dominance [30].

During the early stage of annealing (t ≤ 7.5 min), the rapid increase in cube fraction in the triple-rolled foils indicates that nucleation is primarily driven by gradients in stored energy. The triple-layer configuration introduces a pronounced through-thickness heterogeneity, with higher and spatially varying stored energy near the matte interfaces and within the center layer. This through-thickness energy contrast accelerates recrystallization and promotes extensive cube nucleation.

As annealing proceeds into the intermediate stage (≈7.5–30 min), texture evolution changes from being dominated by nucleation to being controlled mainly by the growth of existing grains. In the triple-rolled foils, this regime is therefore governed by growth competition among the orientations that have already formed during the early stage. On the outer bright and matte surfaces (TR-O-B and TR-O-M), this competition manifests as a transient decrease in cube fraction (Figure 7). EBSD orientation maps and grain-size statistics (Figure 8) show that, in this time interval, residual S- and brass-oriented grains on TR-O-B and TR-O-M coarsen more rapidly and reach larger sizes than the surrounding cube grains. In the central interface layer (TR-C-M), a slight reduction in the average cube grain size is also observed during this stage; however, because the cube component already dominates the texture, its overall cube fraction does not exhibit a pronounced decrease, and instead the increase in cube fraction becomes markedly slower in this time window (Figure 7). This behavior is consistent with orientation-selective boundary migration that temporarily favors the growth of S and brass components at the expense of nearby cube grains on the outer surfaces, leading to the observed dip in cube fraction there [31].

At longer annealing durations (t ≥ 60 min), most deformation-induced substructures have already recrystallized, and further texture evolution is controlled primarily by competitive growth among recrystallized grains. Under these conditions, cube-oriented grains, which are widely recognized as low-energy orientations in recrystallized fcc copper and tend to form energetically favorable boundaries with the surrounding matrix, gradually gain a persistent growth advantage [17]. In the triple-rolled foils, this advantage is most pronounced in the central interface layer (TR-C-M), where cube grains coarsen markedly through subgrain coalescence and high-angle boundary migration, while residual S-, brass-, and copper-type deformation components are progressively fragmented and consumed. As a consequence, the cube texture becomes stabilized across all three layers: cube fractions approach a plateau, and deformation-related orientations are strongly suppressed, resulting in a cube-dominated recrystallization texture in the triple-rolled foils [32].

### 4.3. Correlation Between Microstructure and Flexural Fatigue Performance

#### 4.3.1. Influence of Crystallographic Texture on Flexural Fatigue Behavior

The N_f_ of rolled copper foils is fundamentally determined by how the microstructure accommodates cyclic strain through coordinated elastic and plastic responses. During repeated bending, maintaining uniform strain distribution across neighboring grains is essential to suppress local stress concentration and delay crack initiation.

In copper foils enriched in cube texture ({001}<100>), several {111}<110> slip systems tend to attain comparable Schmid factors under a near-surface plane-stress state during bending [33]. As a result, multiple slip systems can be activated concurrently rather than being funneled into a single family. This promotes more uniform intragranular glide and facilitates early dislocation rearrangement and patterning. In turn, slip compatibility between neighboring grains is improved, elastic–plastic mismatch and boundary tractions are reduced, and boundary-governed crack nucleation is delayed under high-cycle, small-amplitude bending [34,35] (Figure 12).

These microstructural characteristics are consistent with the empirical Coffin–Manson relationship, which describes the correlation between strain amplitude (Δε/2) and fatigue life (N_f_):
(5)Δε2=σ′fE2Nfb+ε′f2Nfc where σ′_f_ and ε′_f_ are the fatigue strength and fatigue ductility coefficients, E is the Young’s modulus, and b (≈−0.05 to −0.12) and c (≈−0.5 to −0.7) are the elastic and plastic fatigue exponents typical for ductile FCC metals such as copper [37]. Within this framework, crystallographic texture modulates the elastic response via the orientation dependence of Young’s modulus. Cube-oriented grains ({001}<100>) exhibit a markedly lower stiffness (E<100> ≈ 66–80 GPa) than {111} grains (E<111> ≈ 180–200 GPa). Figure 13 plots the orientation-averaged Young’s modulus along the bending direction against the flexural fatigue life N_f_. The orientation-averaged Young’s modulus along the bending direction was calculated from the measured ODFs using the single-crystal elastic constants of copper. A clear tendency is observed that specimens with a lower effective modulus generally show a longer flexural fatigue life. Under curvature-controlled bending, a lower E reduces the elastic stress amplitude and the elastic strain-energy density carried by cube grains. This, in turn, diminishes elastic mismatch and grain-boundary tractions, promotes more compatible intergranular deformation, and facilitates more continuous slip transfer during cycling. A similar trend has been reported for electroless Ni–P coatings, where nanoscratching tests showed that the wear resistance was significantly improved with an increasing ratio of hardness (H) to elastic modulus (E), i.e., with a lower effective E at a given hardness [38]. Overall, these results indicate that texture-induced elastic anisotropy makes a significant contribution to N_f_.

Overall, a higher fraction of cube-oriented ({001}<100>) grains is associated with improved flexural-fatigue resistance, owing to their lower orientation-dependent stiffness and more favorable multi-slip coordination under cyclic bending.

#### 4.3.2. Grain-Size Window for Cyclic Plasticity

Grain size governs how plastic strain is partitioned at the microscale. In the 24 µm-thick Cu foils tested under the present flexural fatigue test conditions, at the ultrafine limit (≈1 µm), a boundary-dominated regime tends to prevail: slip paths are truncated and strain becomes discontinuous, leading to poor flexural fatigue resistance [39]. When grains are too fine (<30 µm), the high boundary density restricts dislocation glide and promotes strain incompatibility, accelerating crack initiation. With moderate grain sizes (≈30–60 µm), slip is more continuous and strain is more uniformly partitioned among grains, which corresponds to higher fatigue life [40]. This improvement is associated with a shift from grain-boundary-controlled toward dislocation-mediated deformation, facilitating slip transfer and reducing intergranular stress accumulation [41]. However, when grains exceed ~60 µm, insufficient boundary constraint allows long slip bands to develop, causing strain localization and promoting premature failure.

Overall, within the present experimental window, the effect of grain size on flexural fatigue is strongly window-dependent. Ultrafine grains are over-constrained by grain boundaries, whereas coarse grains are under-constrained by long slip bands; only an intermediate grain-size range stabilizes cyclic plasticity and enhances flexural fatigue performance.

#### 4.3.3. Hetero-Deformation-Induced Strengthening in Partially Recrystallized Microstructures

As shown in Figure 6d, the TR-C-M foil annealed for 7.5 min exhibits pronounced heterogeneity in orientation and grain size within the center layer. Large cube-oriented recrystallized grains coexist with regions that retain finer S- and brass-oriented deformation grains, forming a locally heterogeneous microstructure. During cyclic bending, interfacial strain mismatch between cube-enriched, moderate-grain regions and adjacent deformation-texture zones generates gradients of geometrically necessary dislocations and the associated long-range back stresses (HDI stresses), giving rise to kinematic-hardening-like responses. These internal stresses, together with cross-interface slip and small grain rotations near heterogeneous boundaries, help redistribute plastic strain and delay its local accumulation, thereby stabilizing cyclic plasticity [42].

The post-bending TEM image (Figure 10d) reveals discrete intragranular slip traces within large recrystallized grains, indicating that these grains actively participate in accommodating cyclic strain. In microstructures where such large recrystallized grains coexist with smaller grains of different orientations, the interfaces between them are expected to mediate strain transfer and promote more coordinated deformation among neighboring grains [43]. Accordingly, the TR-C-M specimen annealed for 7.5 min reaches the order of 8.0 × 10^4^ cycles (Figure 9a), consistent with HDI-type strengthening under a cube-enriched condition and moderate grain coarsening, which together contribute to enhanced flexural fatigue resistance.

## 5. Conclusions

Triple-layer stacked rolling establishes pronounced through-thickness strain gradients at interfaces, generating distinct deformation structures across the bright, matte, and central-interface surfaces. During annealing at 600 °C, early-stage recrystallization proceeds rapidly within ~7.5 min, followed thereafter by orientation-selective grain growth. The heterogeneous yet balanced stored-energy distribution is associated with accelerated cube fraction development throughout the multilayered structure. Grain size evolution is layer-dependent, showing moderate coarsening at the outer surfaces and cube-enriched growth in the center layer.

The flexural-fatigue life increases with both a higher cube fraction and the maintenance of a moderate grain size (≈30–60 µm), where intergranular slip compatibility and boundary constraint are better balanced. The central interface (TR-C-M) attains ~8.0 × 10^4^ cycles, consistent with a microstructure combining compliant (100) orientations and coordinated plastic deformation.

Overall, these results support a processing–microstructure–property linkage and indicate that flexural-fatigue resistance can be effectively improved by tailoring grain size and cube texture, providing a microstructural reference for the design of ultrathin copper foils in flexible-electronics architectures.

## Figures and Tables

**Figure 1 nanomaterials-16-00011-f001:**
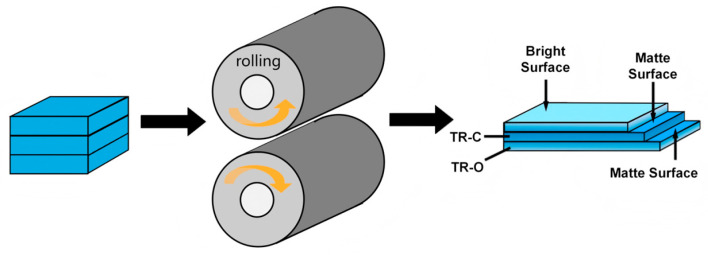
Schematic diagram of the rolling process.

**Figure 2 nanomaterials-16-00011-f002:**
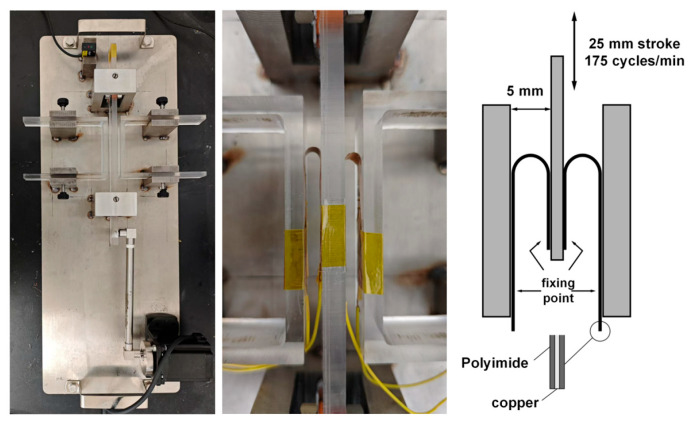
Schematic diagram of cyclic bending test.

**Figure 3 nanomaterials-16-00011-f003:**
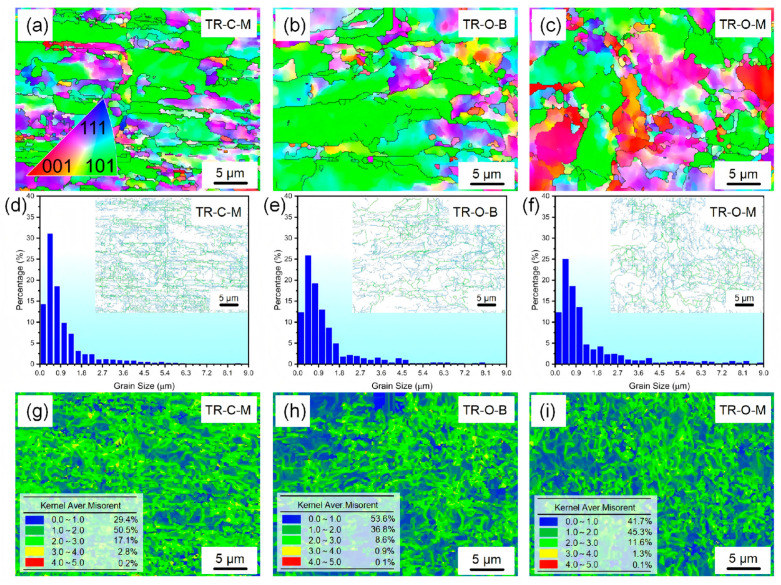
EBSD inverse pole figure (IPF) maps, grain size distributions, and kernel average misorientation (KAM) maps of the cold-rolled triple-rolled (TR) copper foils at different layer positions: (**a**,**d**,**g**) TR-C-M, (**b**,**e**,**h**) TR-O-B, and (**c**,**f**,**i**) TR-O-M. Black lines in IPF maps denote high-angle grain boundaries (>15°). Insets in (**d**–**f**) show boundary maps where blue and green lines represent low-angle (2–15°) and high-angle (>15°) boundaries, respectively. The tables in (**g**–**i**) summarize the quantitative KAM fractions corresponding to the color scales.

**Figure 4 nanomaterials-16-00011-f004:**
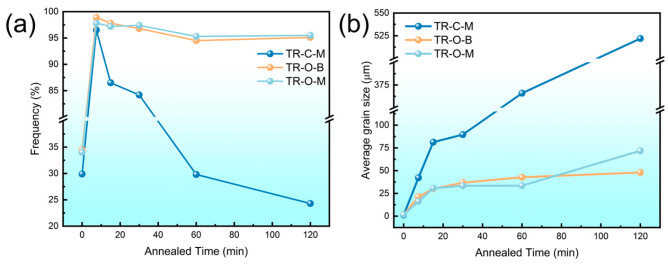
Microstructure of the Cu foils before and after annealing: (**a**) HAGB fractions; (**b**) average grain size.

**Figure 5 nanomaterials-16-00011-f005:**
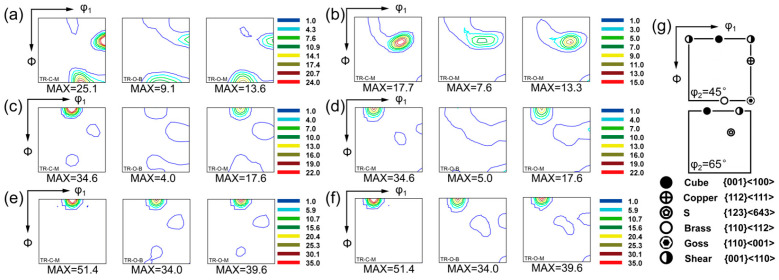
ODF sections of the Cu foils at the rolled and annealed states: (**a**) rolled state with φ2 = 45°, (**b**) rolled state with φ2 = 65°, (**c**) annealed at 600 °C for 7.5 min with φ2 = 45°, (**d**) annealed at 600 °C for 7.5 min with φ2 = 65°. (**e**) annealed at 600 °C for 2 h with φ2 = 45°, (**f**) annealed at 600 °C for 2 h with φ2 = 65°. (**g**) Schematic illustration of the ideal texture components for Cu.

**Figure 6 nanomaterials-16-00011-f006:**
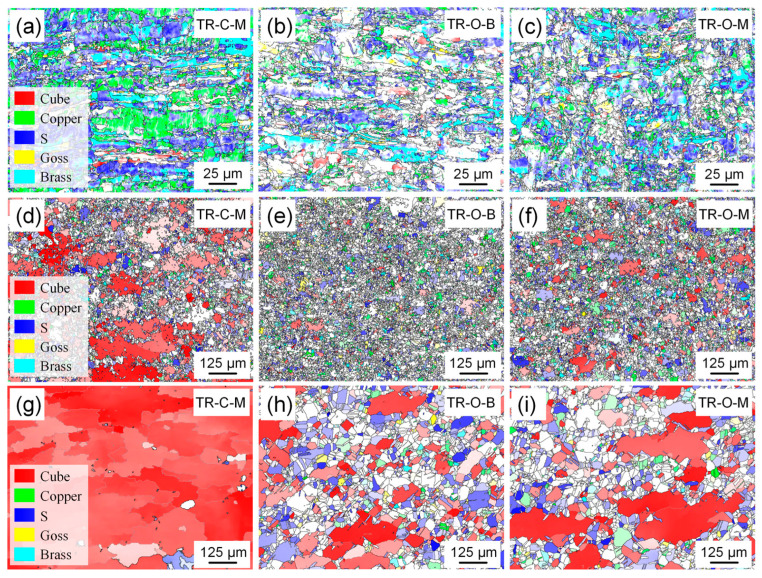
Distribution maps of the selected texture components (cube, copper, s, goss, and brass) on different surfaces of the triple-rolled Cu foil: (**a**–**c**) rolled state, (**d**–**f**) annealed at 600 °C × 7.5 min, and (**g**–**i**) annealed at 600 °C × 120 min. (**a**,**d**,**g**) correspond to the center interface surface (TR-C-M), (**b**,**e**,**h**) to the outer bright surface (TR-O-B), and (**c**,**f**,**i**) to the outer matte surface (TR-O-M).

**Figure 7 nanomaterials-16-00011-f007:**
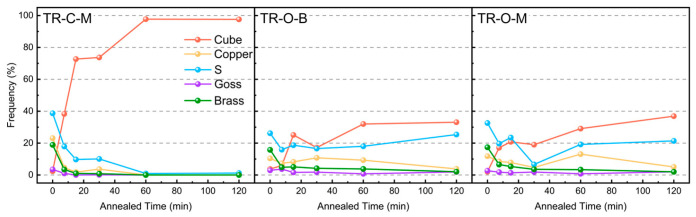
Fractions of texture components in the rolled and annealed Cu foils.

**Figure 8 nanomaterials-16-00011-f008:**
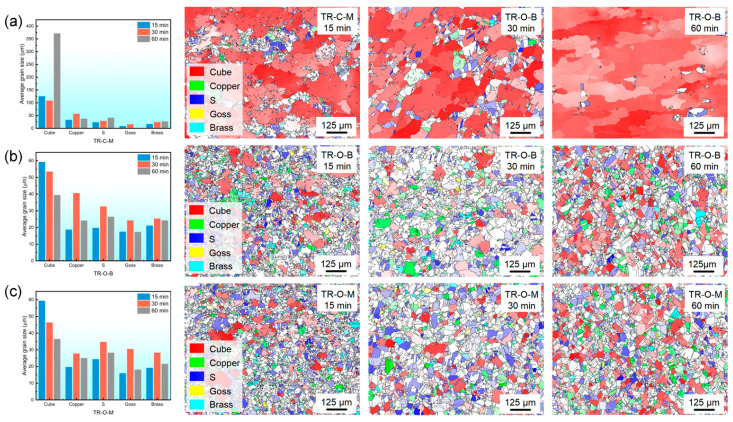
Evolution of average grain size for different texture components and corresponding orientation maps of (**a**) TR-C-M, (**b**) TR-O-B, and (**c**) TR-O-M foils annealed at 600 °C for 15, 30, and 60 min.

**Figure 9 nanomaterials-16-00011-f009:**
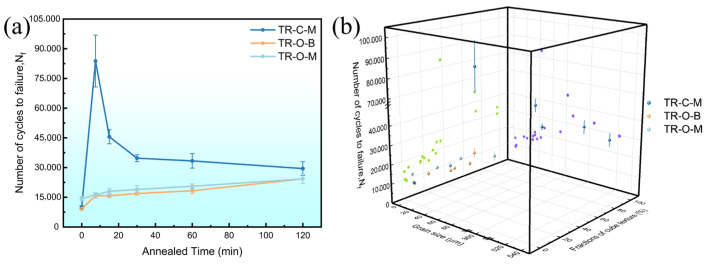
(**a**) Flexural fatigue life (N_f_) versus annealing time, each surface tested as the inner bending surface. (**b**) 3D correlation among cube fraction, grain size, and N_f_; each point pairs cyclic-bending data with EBSD-derived cube fraction and grain size.

**Figure 10 nanomaterials-16-00011-f010:**
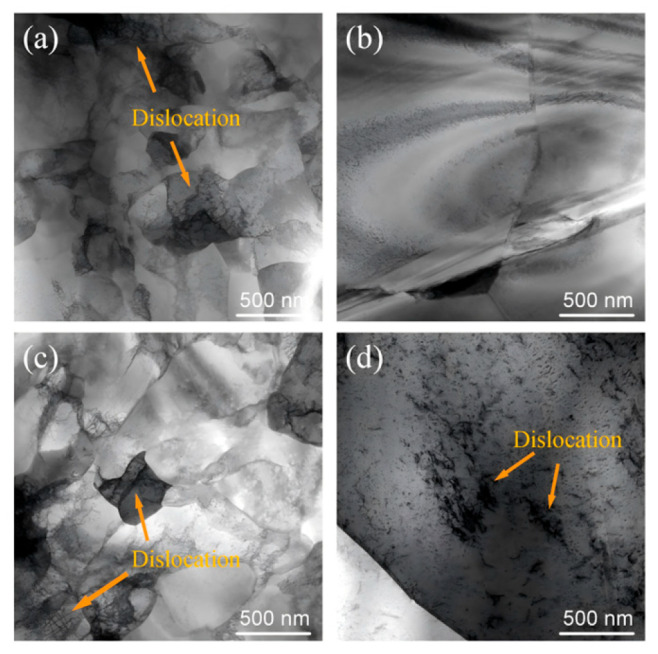
Bright-field TEM images of TR-C-M specimens before bending and after cyclic bending fracture: (**a**,**c**) cold-rolled state and (**b**,**d**) annealed state (7.5 min); (**a**,**b**) pre-bending regions and (**c**,**d**) post-fracture regions.

**Figure 11 nanomaterials-16-00011-f011:**
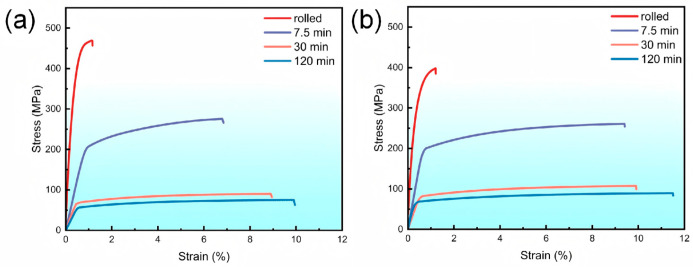
Tensile stress–strain curves of triple-rolled Cu foils annealed at 600 °C for different durations: (**a**) TR-C and (**b**) TR-O.

**Figure 12 nanomaterials-16-00011-f012:**
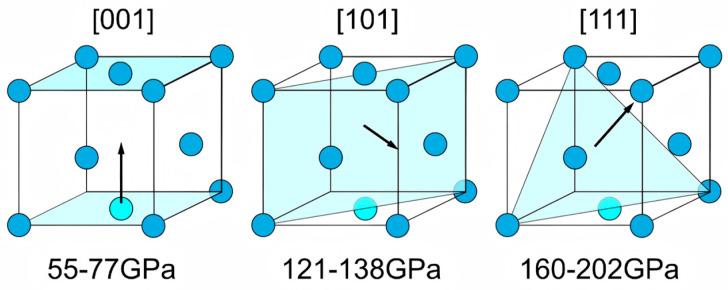
Anisotropy of the Young’s modulus on copper crystal, the arrows denote the loading direction ([001], [101], and [111]), and the shaded planes indicate the corresponding planes normal to the loading direction: (001), (101), and (111), respectively [36].

**Figure 13 nanomaterials-16-00011-f013:**
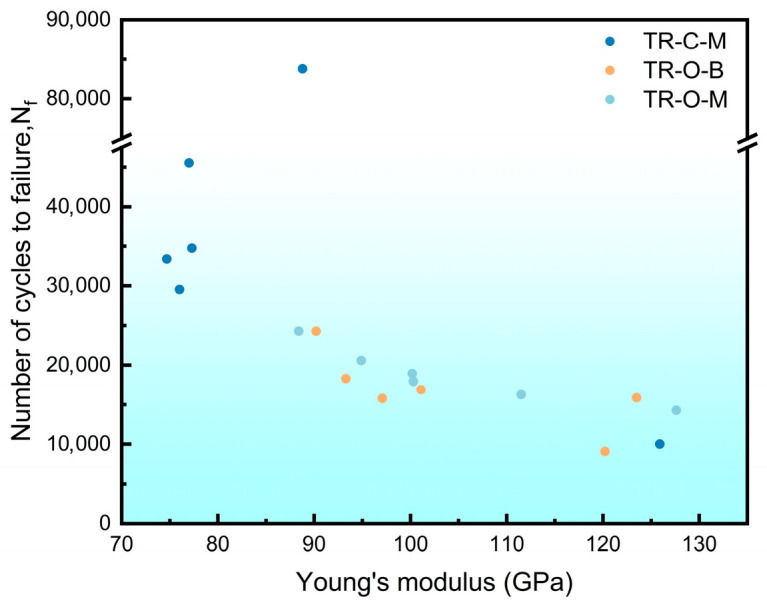
Scatter plot of Young’s modulus (E) versus flexural fatigue life (N_f_).

## Data Availability

The original contributions presented in this study are included in the article/Appendix A. Further inquiries can be directed to the corresponding author.

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
