# Peer review of "Texture and Flexural Fatigue Resistance Governed by Surface-Dependent Deformation and Recrystallization in the Copper Foils"

_nanomaterials, 2025, doi:10.3390/nano16010011_

Round 1
Reviewer 1 Report
Comments and Suggestions for Authors
- The abstract is informative but remains too general; it should more clearly specify the methodological details and the precise novelty of the study to help readers understand its unique contribution.
- The fatigue failure criterion based on electrical interruption requires validation. Please mention whether fracture surfaces were inspected
- Justify the choice of 15° as the critical misorientation angle for HAGBs; some works use 10–15°.
- The decrease and later increase of cube fraction in TR-O-B and TR-O-M (Fig. 7) suggests orientation competition, but no deeper mechanistic analysis is provided.
- Fig. 4(a): The rapid decrease in HAGB fraction at long annealing times needs further explanation (e.g., whether this is due to AGG or orientation sharpening).
- Improve clarity and contrast in Figures 5–7.
- Clarify the repeatability of fatigue tests (number of samples, variation).
- If possible make the long long sentences in the discussion section for readability. It will make easy to read.
- The conclusions are consistent with the study, but please avoid overstating claims about industrial relevance unless benchmarked against flexible-electronics fatigue requirements.
Reviewer 2 Report
Comments and Suggestions for Authors
Overall, it is a good piece of work. I have several minor comments:
1. The statistics of fatigue testing need to be strengthened. Only 3 specimens per condition have been included. More repeated testing is needed, with proper error bars for each mechanical testing result figure. Consider providing raw data as supplementary materials in the paper.
2. The definition of triple-rolled geometry in the implemented bending tests needs to be clearer. How is TR-C-M ensured in the tension side during the cyclic bending? For EBSD/TEM, which sides are you mapping? (The tensile surface/compressive surface/mid-thickness plane?)
3. More clarifications of EBSD measurement details, such as step size, indexing parameters, etc.
4. Add GND densities based on KAM maps --> for stored energy argument.
5. For such a thin layer of large grain size (after annealing), the quantification of HAGB is noisy and would be very sensitive to misorientation threshold. Clarify this point and modify the interpretation based on these results, such as cube clustering.
6. Section 4.3.1 Can the author compute the orientation-averaged Young’s modulus along the bending direction based on measured ODFs and plot against Nf to strengthen the argument? Each other's contribution to Nf (such as grain size) can therefore be quantified and compared.
Round 2
Reviewer 2 Report
Comments and Suggestions for Authors
A couple of more comments:
1. Line 301: check reference.
2. No table S1-S3 (raw data of fatigue life) is included in the manuscript. The authors should either (1) present all five data points with standard deviation/error bars, or (2) justify the exclusion of extrema using a recognized statistical criterion (e.g., Grubbs’ test). At minimum, Fig. 9 should include error bars or scatter markers to convey data variability.
3. Line 177: GND density reflects plastic strain gradients, which are associated with the elastic energy stored in the dislocation stress field, rather than the stored elastic energy as claimed by the author.
Author Response
Comment 1: Line 301: check reference.
Response 1: Thank you for the comment. We checked Line 303 and found that a supporting reference was missing. We have added Ref. 20 to substantiate the statement (Sutou, Y.; Omori, T.; Kainuma, R.; Ishida, K. Acta Materialia 2013, 61, 3842–3850; DOI: 10.1016/j.actamat.2013.03.022).
Comment 2: No table S1-S3 (raw data of fatigue life) is included in the manuscript. The authors should either (1) present all five data points with standard deviation/error bars, or (2) justify the exclusion of extrema using a recognized statistical criterion (e.g., Grubbs’ test). At minimum, Fig. 9 should include error bars or scatter markers to convey data variability.
Response 2: Thank you for this suggestion. We have revised the fatigue-life definition and data presentation to explicitly convey data variability. Specifically, in the Methods section (Page 3, Lines 105–107), we now define the flexural fatigue life as the arithmetic mean of the number of cycles to failure obtained from all five specimens (n = 5) for each condition, and we state that the raw fatigue-life data are provided in Tables S1–S3. In addition, we have re-plotted Fig. 9 and Fig. 13 using the updated dataset, and added error bars (±SD, n = 5) to Fig. 9 to reflect the variability among the five measurements.
Comment 3: Line 177: GND density reflects plastic strain gradients, which are associated with the elastic energy stored in the dislocation stress field, rather than the stored elastic energy as claimed by the author.
Response 3:
Thank you for pointing this out. We have revised the manuscript accordingly.We have revised the text (Page 5–6, Lines 186–190) to state that surface-dependent differences in GND density reflect through-thickness plastic strain gradients, which are associated with the elastic energy stored in the dislocation stress field, and we removed the previous wording that directly linked GND density to “stored elastic energy.”